# Call it a conspiracy: How conspiracy belief predicts recognition of conspiracy theories

J. P. Prims [ID] *

Department of Psychology, University of Illinois at Chicago, Chicago, Illinois, United States of America

* jprims2@uic.edu

**Data Availability Statement:** All data and analysis files are available on OSF (https://osf.io/h3gdr/?view_only=c092855b387243269fc3ac9751b4fff9).

**Funding:** The author(s) received no specific funding for this work.

## Abstract

While conspiracy theories are treated as irrational fringe beliefs in popular culture, conspiracy belief is quite common. Given the disconnect between stereotypes about conspiracy belief and its prevalence, I tested whether people have difficulty recognizing the conspiracy theories that they believe as conspiracy theories. Across two studies I demonstrate that people have considerable difficulty identifying conspiracy theories they believe as conspiracy theories, particularly when they do not take much time to consider whether their beliefs might be conspiracy theories. This is consistent with the notion that people experience "conspiracy blindness." People have trouble recognizing the conspiracy theories they believe as conspiracy theories because they do not take the time to consider whether their beliefs might be conspiracy theories. In Study 2, I demonstrate that people can overcome their conspiracy blindness and recognize the conspiracy theories they believe as conspiracy theories when they are given a definition for "conspiracy theory" and asked to consider their answer. This suggests that people are typically ignorant of their own conspiracy beliefs, but capable of recognizing them when given the tools and motivation to do so. However, recognizing their beliefs as conspiracy theories does not reduce their adherence to those beliefs.

## Introduction

Conspiracy theorists do not have the best public image. Late night talk show hosts mock their intelligence and sanity [1,2], popular media portrays them as isolated eccentrics [3–5], and stereotypical images portray them with tinfoil hats. It is little wonder that people who believe conspiracy theories reject the stigmatized labels of "conspiracy theory" "conspiracy theorist" [6,7].

In academic circles, conspiracy theories are explanations for events or circumstances that claim a group of powerful people is working together to accomplish a goal that comes at the expense of others, while attempting to keep their actions or intentions a secret [8–10]. There is considerable debate within academia over whether conspiracy theories are inherently irrational [11]. After all, real conspiracies do occur (e.g., Watergate and MK Ultra). This debate does not seem to have spread very far beyond academia. Conspiracy theories and people who believe them are often considered irrational by lay people [7,12] and, on occasion, by the people who study them (as observed and critiqued by [13,14]).

Despite the stigma surrounding it, belief in conspiracy theories (i.e., "conspiracy belief") is widespread. Representative studies show that over 50% of U.S. [15] and Croatian [16] citizens

**Competing interests:** The authors have declared that no competing interests exist.

were willing to endorse at least one conspiracy theory from a curated list. An exhaustive list of theories would likely yield a higher percentage. For example, the percentage of Americans that believe conspiracy theories about the Kennedy assassination has ranged between 50 and 81% since 1963 [17].

Belief in conspiracy theories is associated with a variety of positive and negative outcomes. As previously noted, real conspiracies do occur. If not for conspiracy theories and the people who believe them, the Nixon campaign's political espionage may have gone unchecked. Belief in conspiracy theories may also be associated with a variety of psychological benefits including entertainment and bolstering egos, [18]. In this sense, belief in conspiracy theories has notable positive effects on individuals and society.

However, belief in conspiracy theories is correlated with risky health behaviors [19–22], neglect of environmental issues [23], and decreased political participation [21]. In addition, there is preliminary evidence that belief in conspiracy theories may be used to justify political violence (i.e., the January 6th Insurrection [24]). Given their association with a variety of positive and negative outcomes, it is important that we understand when, where, and why people believe conspiracy theories, and why belief in conspiracy theories is so common despite the stigma surrounding it.

Preliminary evidence suggests that many people who believe conspiracy theories do not label the things they believe as conspiracy theories [25]. This may explain why so many people believe conspiracy theories despite the stigma: They do not realize or acknowledge that their beliefs are conspiracy theories.

I propose two explanations for this association between belief and inability (or unwillingness) to recognize conspiracy theories: First, it may be that these people recognize that others may consider their beliefs "conspiracy theories," but are motivated to reject that label due to the perceptions of irrationality and stigma surrounding belief in conspiracy theories. I will refer to this explanation as the "motivated reasoning hypothesis." Second, it may never cross peoples' minds that their beliefs might be conspiracy theories; they are simply blind to the possibility. I will refer to this as the "conspiracy blindness hypothesis."

## Motivated reasoning hypothesis

Motivated reasoning is a process through which pre-existing biases affect reasoning [26]. When people engage in motivated reasoning, they seek out arguments and information that support their pre-existing beliefs and ignore or discredit arguments and information that does not support their pre-existing beliefs. Conspiracy beliefs themselves are facilitated by motivated reasoning [27,28]. Motivated reasoning may influence not just conspiracy belief but may also hamper believers' ability to acknowledge that some of their beliefs qualify as conspiracy theories.

Self-assessments of knowledge and cognitive ability are prone to bias. The average person believes that they are relatively knowledgeable, intelligent, and rational compared to (their perception of) the average person [29–32]. The average person also believes that conspiracy theories are "irrational" or "foolish" [33,34]. However, the majority of the U.S. population believes in at least one conspiracy theory. Given that people are biased to believe that they themselves are rational, and that many people believe that conspiracy theories are irrational, people who believe conspiracy theories may engage in motivated reasoning to conclude that they are rational and do not believe "irrational" conspiracy theories.

People may also be motivated to avoid associations with stigmatized labels. The labels "conspiracy theory" and "conspiracy theorist" are associated with social stigma [33–35]. People may not wish to label their beliefs as conspiracy theories to avoid the social stigma associated with that label.

The motivated reasoning hypothesis predicts that when people are presented with a conspiracy theory that they believe, they will be reluctant to label it a "conspiracy theory" because they are motivated to conclude that their beliefs are rational (i.e., not "irrational" conspiracy theories) or avoid the social stigma associated with believing "conspiracy theories."

## Conspiracy blindness hypothesis

Most people rarely have the cognitive resources or motivation to make careful, well-researched, and informed decisions. During initial information processing, they may rely on heuristics (i.e., "cognitive shortcuts") to judge the accuracy of information and beliefs. These heuristics bypass in-depth evaluation of information and may lead to incorrect or incomplete conclusions. When relying on heuristics, people may fail to evaluate the validity of their own arguments [36] accept politically congenial information as the truth without evaluating it [37], or fail to evaluate whether a news story is true or false, provided that the headline seems plausible [38].

The conspiracy blindness hypothesis predicts that, when people are presented with a conspiracy theory that they believe, they will have difficulty correctly labeling it as a "conspiracy theory" because they will not think to consider that it might be a conspiracy theory. The (mistaken) belief that conspiracy theories are inherently false may preclude them from even considering that something they think is plausible could possibly be a conspiracy theory.

Neither of these hypotheses predict that awareness that one's own beliefs are conspiracy theories would reduce adherence to those beliefs. While labeling a belief a "conspiracy theory" may reduce willingness to express that belief aloud, it does not appear to reduce adherence to that belief [12,25,39].

I tested these two hypotheses in two studies. In Study 1, participants read summaries of twenty real news articles: Ten articles did not contain conspiracy theories (i.e., "mainstream articles") and ten did contain conspiracy theories (i.e., "conspiracy articles"). Participants rated how true they thought each article was as a measure of belief. They then decided whether the article contained a conspiracy theory. I timed how long participants took to decide whether the article contained a conspiracy theory. If the motivated reasoning hypothesis is true, then participants will take additional time to reason out why the things they believe "do not count" as conspiracy theories. As a result, they will be more likely to (slowly) incorrectly categorize the conspiracy articles they believe as mainstream articles. However, if the conspiracy blindness hypothesis is true, then participants will not consider whether the conspiracy theories they believe are conspiracy theories and make a snap decision without attempting to justify their decision. As a result, they will be more likely to (quickly) incorrectly classify the conspiracy articles they believe as mainstream.

In Study 2, I attempted to replicate the results of Study 1 using short statements intentionally written to contain or not contain conspiracy theories. As before, if the motivated reasoning hypothesis is true, then participants will be motivated to take additional time to reason that the conspiracy theories they believe "do not count" as conspiracy theories. As a result, they will be more likely to (slowly) incorrectly categorize the conspiracy statements they believe as mainstream statements. As before, if the conspiracy blindness hypothesis is true, then participants will not consider whether the conspiracy theories they believe are conspiracy theories and decide quickly without bothering to think about their decision, and as a result, will be more likely to incorrectly classify the conspiracy statements they believe as mainstream statements.

I also tested whether asking participants to apply the definition of the term "conspiracy theory" to the statements improved their accuracy in identifying conspiracy theories that they

believed. If the motivated reasoning hypothesis is true, giving participants the definition for "conspiracy theory" should have little to no impact on their accuracy in correctly identifying conspiracy theories, because they are motivated to believe that their beliefs are not "irrational" or stigmatized "conspiracy theories." If the conspiracy blindness hypothesis is true, giving participants the definition for "conspiracy theory" should improve their accuracy in correctly identifying all conspiracy theories regardless of their beliefs. By asking them to apply the definition for "conspiracy theory" to each statement, participants are encouraged to consider whether a statement could be a conspiracy theory when they had not considered it before.

## Study 1

Study 1 served as an initial test of the motivated reasoning and conspiracy blindness hypotheses. In this study, participants viewed summaries of 20 news articles (pre-selected to contain or not contain conspiracy theories) and then decided whether they thought that the article summary contained or did not contain a conspiracy theory.

## Method

### Participants

Two hundred and fifty four participants from Amazon Mechanical Turk (mTurk) completed the survey between September 14th and September 15th, 2020. This sample size is sufficient for more than 80% power with an alpha of .05, based on a Monte Carlo simulation ($N = 200$ samples) with an estimated effect size of Cohen's $d = .50$ using the simr package (version 1.0.5 [40]). This estimated effect size is based on the results of two similar studies that examined the effect of information processing (i.e., how critically participants thought about their answers) and ideological congeniality (i.e., how consistent the information was with participants' pre-existing beliefs) on participants' ability to identify fake news [38]. This sample size was decided upon before data collection began, and data collection ended before analysis began.

The average age of the sample was 34.96 ($SD = 9.58$). One-hundred and sixty-four participants identified as men, 89 identified as women, and one identified as nonbinary. The majority of the sample ($n = 143$) had a four-year college degree. Forty-seven had a professional degree, 29 had some college, 15 had a two-year degree, 14 were high school graduates, five had a doctorate, and one had less than a high school degree. One-hundred and seventy-three participants identified as White, 44 identified as Black or African American, 22 identified as Asian, 8 identified as Hispanic or Latino, two identified as something else, one identified as American Indian or Alaska Native, and four identified as multiracial. On a scale of -3 (Very Liberal) to 3 (Very Conservative) the sample averaged 0.21 ($SD = 2.01$). Ninety-eight participants identified on the liberal side of the spectrum, 122 identified on the conservative side of the spectrum, and 34 identified as moderate.

All participants provided informed consent via an electronic consent form. This research was approved by the University of Illinois at Chicago's Institutional Review Board under protocol #2020–1103.

### Procedure

After a brief introduction to the survey, participants read twenty article summaries. The selection criteria for the articles in Study 1 and statements in Study 2, as well as the full headlines, statements, and summaries, are presented in the pilot study reported in the supplemental material. Ten of these article summaries were from mainstream news sites (mainstream articles) and ten were from conspiracy news sites (conspiracy articles). None of the mainstream

articles contained conspiracy theories, and all the conspiracy articles contained at least one conspiracy theory. The conspiracy articles contained conspiracy theories that fit the definition used in this paper. They did not have to be proven false to be included, but none were considered "true" by mainstream narratives at the time of data collection.

All article summaries are available in S1 Table. To measure how much participants believed each article summary, participants rated each article summary on a scale of 1 (Completely False) to 7 (Completely True). Immediately after reporting their belief in an article summary, participants decided whether the article contained a conspiracy theory or did not contain a conspiracy theory The number of seconds they took to make their decision (their response time) served as a proxy measure of how much they thought about their decision. Shorter response times are associated with more surface-level information processing, whereas longer response times are associated with deeper levels of information processing [41]. Given this association between response time and information processing, longer response times have been used to measure motivated reasoning in past research (e.g., [42–44]).

I measured response time using Qualtrics's "Last click" timing data. This feature measures the number of seconds from the time the participant loads the survey page until they make their last click on that page (not including when they click to advance to the next page). The question asking whether the article summary was on its own page, so for most cases, the "Last click" timing represents the number of seconds between the page loading and participants clicking on their final answer.

After deciding whether the article summary contained a conspiracy theory, I asked participants whether they would like to write a justification for their decision (yes or no). If they selected 'yes,' participants were given a textbox and asked to explain their reasoning. If they selected 'no,' they moved on to the next article summary. After participants rated and categorized all 20 article summaries, they had the opportunity to write their own definition for the term "conspiracy theory." Finally, participants provided basic demographic information including age, gender, race and ethnicity, political orientation, and educational background. The full survey is available in the supplementary material.

Participants' belief in each article summary was rescored to a scale of -3 (Completely false) to 3 (Completely true) and centered on 0, such that negative scores indicated disbelief and positive scores indicated belief. Participants' political orientation was rescored to a scale of -3 (Very liberal) to 3 (Very conservative) and centered on 0, such that more negative numbers indicated that the participant was more liberal, and more positive scores indicated that the participant was more conservative. I scored each participant's accuracy by determining whether they correctly identified the conspiracy theory (or lack there-of) in each article summary. For example, I coded correctly stating that the conspiracy articles contained a conspiracy theory as correct (1) and incorrectly stating that the conspiracy articles did not contain a conspiracy theory as incorrect (0).

Similar studies have operationalized participants' accuracy using d' (d-prime). d' quantifies accuracy by taking a participant's number of "hits" (i.e., correctly identifying a conspiracy headline as a conspiracy headline) and subtracting their number of "false positives" (i.e., incorrectly identifying a mainstream headline as a conspiracy headline) from a large set of trials (i.e., many conspiracy and mainstream headlines). Although this statistic is beneficial for many types of research, it is not suited for this study. First, d' collapses across stimuli. This means that d' would reduce the predictive power of the model by treating all conspiracy headlines as equally believable for all participants. Because belief in different genres of conspiracy theories can look very different (e.g., political vs apolitical [45]), collapsing across genre would be an inappropriate choice when studying conspiracy theory belief. Second, d' is only a valid measure of detection if the variance in accuracy is the same for both mainstream and

conspiracy headlines. I predicted little variance for mainstream headlines and considerable variance for conspiracy headlines thereby violating this base assumption. Third, d' produces a single number that represents accuracy for a set of 20 statements, while using accuracy allowed me to analyze each summary as a separate observation, greatly increasing my statistical power. For these reasons, I decided to use overall accuracy rather than d' in Studies 1 and 2.

## Results

Means, standard deviations, and correlations between all demographic and key variables are available in Table 1. Means and standard deviations of belief for individual article summaries, and correlations with all demographic and key variables, are available in the supplemental material (S3–S6 Tables).

Both the motivated reasoning hypothesis and the conspiracy blindness hypothesis predict a three-way interaction between belief, article type, and response time. Both predicted that there would not be an interaction between belief and reaction time for mainstream article summaries, but that there would be an interaction between belief and reaction time for conspiracy article summaries.

The motivated reasoning hypothesis predicts that the more people believe a conspiracy article summary, the less accurate they will be when they take longer to decide whether it contains a conspiracy theory (because they are engaging in motivated reasoning). The conspiracy blindness hypothesis predicts that the more people believe a conspiracy article summary, the less accurate they will be when they take shorter to decide whether it contains a conspiracy theory. To test the motivated reasoning and conspiracy blindness hypotheses, I used a logistic mixed-effects model. I took a forward-fitting approach, adding the predictors in stages [46]. I performed all analyses using the lme4 package in R (lme4 ver. 1.1–23 [47]). All models used the same random effects structure. The random effects were crossed. Article summary (item) was nested within article type (mainstream or conspiracy), and article type was nested within subjects. Article summary was treated as a repeated measure for subjects, but a nested variable for article type [48,49]. Article summary and subjects were allowed their own intercepts, and

**Table 1. Means, standard deviations, and correlations between key variables with confidence intervals.**

| Variable | M | SD | 1 | 2 | 3 | 4 | 5 | 6 |
|---|---|---|---|---|---|---|---|---|
| 1. Age | 34.96 | 9.58 | | | | | | |
| 2. Education | 4.76 | 1.13 | -.05 | | | | | |
| | | | [-.17, .07] | | | | | |
| 3. Political Orientation | 0.21 | 2.01 | -.02 | .27** | | | | |
| | | | [-.14, .10] | [.16, .38] | | | | |
| 4. Mainstream Belief | 0.89 | 0.85 | -.06 | .02 | .05 | | | |
| | | | [-.18, .07] | [-.10, .14] | [-.07, .17] | | | |
| 5. Conspiracy Belief | 0.31 | 1.11 | -.24** | .31** | .37** | .39** | | |
| | | | [-.35, -.12] | [.19, .41] | [.26, .47] | [.29, .49] | | |
| 6. Reaction Time (in Seconds) | 4.59 | 4.89 | -.07 | .20** | .12 | .03 | .26** | |
| | | | [-.19, .05] | [.08, .31] | [-.00, .24] | [-.09, .15] | [.14, .37] | |
| 7. Score | 12.85 | 3.95 | .22** | -.40** | -.36** | .10 | -.60** | -.25** |
| | | | [.10, .34] | [-.50, -.29] | [-.46, -.24] | [-.02, .22] | [-.67, -.51] | [-.36, -.13] |

*Note.* M and SD are used to represent mean and standard deviation, respectively. Values in square brackets indicate the 95% confidence interval for each correlation. * indicates $p < .05$. ** indicates $p < .01$. "Mainstream Belief" is the average reported belief for all mainstream articles. "Conspiracy belief" is the average reported belief for all conspiracy items. Score represents the total number of headlines (out of 20) participants correctly classified as "mainstream" or "conspiracy."

article type was entered as a random slope. I blocked the correlation between subjects and article type.

Model 0 contained no fixed effects. Model 1 tested for a relationship between belief (mean centered and z-scored) and accuracy (whether participants correctly identified whether the article was a conspiracy article or a mainstream article). Model 2 included article type (mainstream, coded as -0.5 or conspiracy, coded as 0.5). Model 3 added the interaction between belief and article type. Model 4 added response time (log 10 transformed, mean centered, and z-scored). Model 5 added the three-way interaction between belief, article type, and response time. Model 6 added education (z-scored) and political orientation (centered on the scale's midpoint and standardized) as level two control variables due to their consistent association with conspiracy belief in the literature [50]. Model 6, which included the main effects of belief, article type, and response time, and their interactions and control variables was the best fit. As both the motivated reasoning and conspiracy blindness hypotheses predict a three-way interaction between belief, article type, and response time, I predicted that either Model 5 or 6 should be the best fit model. All model comparisons are available in Table 3, and all model coefficients are available in Table 2.

Even though neither the motivated reasoning nor the conspiracy blindness predicted main effects of belief, article type, or response time, all three were significant. The more participants believed the article summary, the better they were at recognizing whether or not it contained a conspiracy theory. Participants were more accurate when deciding if conspiracy articles contained conspiracy theories than if mainstream articles contained conspiracy theories. Finally, the faster participants made their decision about whether the article contained a conspiracy theory, the more accurate they were.

Consistent with both the motivated reasoning and conspiracy blindness hypotheses, and replicating Douglas, van Prooijen, & Sutton [25], the more participants believed the conspiracy articles, the less likely they were to recognize that they contained conspiracy theories ($\beta$ = -0.75, $SE$ = 0.07, $p < .001$, $OR$ = 0.47, Cohen's $d$ = -0.42). Although neither the motivated reasoning nor the conspiracy blindness hypotheses made specific predictions about the belief and accuracy for mainstream articles, the more participants believed the mainstream articles, the more likely they were to correctly recognize that they did not contain conspiracy theories, $\beta$ = 0.33, $SE$ = 0.08, $p < .001$, $OR$ = 1.39, Cohen's $d$ = 0.18. Overall, belief's relationship with article classification accuracy went in opposite directions for conspiracy and mainstream articles. Belief was associated with less accuracy for conspiracy articles (i.e., thinking conspiracy articles did not contain conspiracy theories), and with more accuracy for mainstream articles (i.e., recognizing that mainstream articles did not contain conspiracy theories).

**Table 3. Summary of comparisons between models, testing whether adding terms improved model fit.**

| Model Comparison | Change in *df* | Deviance | $\chi^2$ | *p* |
|---|---|---|---|---|
| *Model 1 versus Model 0 (Null)*<br>Belief | +1 | 4550.45 | 44.92 | < .001 |
| *Model 2 versus Model 1*<br>Belief + Article Type | +1 | 4544.49 | 5.95 | .015 |
| *Model 3 versus Model 2*<br>Belief * Article Type | +1 | 4443.02 | 101.46 | < .001 |
| *Model 4 versus Model 3*<br>Belief * Article Type + Response Time | +1 | 4431.29 | 11.73 | < .001 |
| *Model 5 versus Model 4*<br>Belief * Article Type * Response Time | +3 | 4410.58 | 20.71 | < .001 |
| *Model 6 versus Model 5*<br>Belief * Article Type * Response Time + Education + Political Orientation | +2 | 4354.47 | 56.10 | < .001 |

**Table 2. Multilevel model results predicting the likelihood of a correct answer from belief, article type, and response time.**

| | Model 1 | Model 2 | Model 3 | Model 4 | Model 5 | Model 6 |
|---|---|---|---|---|---|---|
| (Intercept) | 1.15*** (0.12) | 1.15 (0.12)*** | 1.04 (0.11)*** | 1.04 (0.10)*** | 1.03 (0.10)*** | 1.03 (0.10)*** |
| Belief | -0.36*** (0.05) | -0.35 (0.05)*** | -0.24 (0.06)*** | -0.25 (0.06)*** | -0.24 (0.06)*** | -0.21 (0.06)*** |
| Article Type | | 0.86 (0.35)* | 0.94 (0.35)** | 0.93 (0.34)** | 0.92 (0.34)** | 0.83 (0.34)* |
| Belief * Article Type | | | -1.15 (0.12)*** | -1.14 (0.12)*** | -1.09 (0.12)*** | -1.07 (0.12)*** |
| Response time | | | | -0.17 (0.05)*** | -0.16 (0.05)** | -0.14 (0.05)** |
| Belief * Response time | | | | | 0.11 (0.05)* | 0.12 (0.05)* |
| Article Type * Response time | | | | | 0.35 (0.10)*** | 0.34 (0.10)** |
| Belief * Article Type * Response time | | | | | 0.23 (0.10)* | 0.23 (0.10)* |
| Education | | | | | | -0.50 (0.09)*** |
| Political Orientation | | | | | | -0.32 (0.09)*** |
| AIC | 4560.45 | 4556.49 | 4457.02 | 4447.29 | 4432.55 | 4380.48 |
| BIC | 4593.11 | 4595.69 | 4502.76 | 4499.56 | 4504.42 | 4465.41 |
| Log Likelihood | -2275.22 | -2272.25 | -2221.51 | -2215.65 | -2205.28 | -2177.24 |
| N | 5080 | 5080 | 5080 | 5080 | 5080 | 5080 |
| **Random Effects** | | | | | | |
| Intercept | 1.69 | 1.66 | 1.36 | 1.26 | 1.26 | 0.94 |
| Article type (Intercept) | 23.73 | 23.36 | 24.27 | 24.05 | 23.74 | 23.90 |
| Item | 0.09 | 0.08 | 0.04 | 0.04 | 0.04 | 0.05 |

*Note.* * indicates a $p$-value that is less than .05

** indicates a $p$-value that is less than .01, and

*** indicates a $p$-value that is less than .001. Standard errors are presented in parentheses. $p$-values are based on asymptotic Wald tests (lme4 ver. 1.1–23; [47]). This analysis did not exclude responses with abnormally long response times. The maximum response time for any item for any participant was 483.02 seconds. A later analysis that excluded items with response times three standard deviations above the mean did not change the pattern of results, though the three-way interaction became marginally significant ($p$ = .084) and the interaction between belief and response time was no longer significant ($p$ = .729).

The amount of time participants took to decide whether an article contained a conspiracy theory did not predict accuracy for conspiracy articles ($\beta$ = 0.02, $SE$ = 0.07, $p$ = .718, $OR$ = 1.02, Cohen's $d$ = 0.01), but faster decisions predicted more accuracy when evaluating mainstream statements, $\beta$ = -0.31, $SE$ = 0.07, $p <$ .001, $OR$ = 0.73, Cohen's $d$ = -0.17. In other words, the main effect of response time on belief is likely driven by the belief in mainstream articles and not conspiracy articles. This finding is inconsistent with the motivated reasoning and conspiracy blindness hypotheses that there would no relationship between response time and accuracy for mainstream headlines and a relationship (negative for the motivated reasoning hypothesis or positive for the conspiracy blindness hypothesis) between response time and accuracy for conspiracy headlines.

The more participants believed the article summary, the less accurate they were when they took less time to respond (1 SD below the mean; $\beta$ = -0.33, $SE$ = 0.07, $p <$ .001, $OR$ = 0.72, Cohen's $d$ = 0.18). There was no relationship between belief and accuracy when they took more time to respond (1 SD above the mean; $\beta$ = -0.09, $SE$ = 0.08, $p$ = .221, $OR$ = 0.91, Cohen's $d$ = -0.05). In other words, how much participants believed the article only predicted their accuracy (higher belief was associated with less accuracy) when they were making quick decisions. Although neither the motivated reasoning hypothesis nor the conspiracy blindness hypotheses made explicit predictions about how belief and response time predicted accuracy, this result is more consistent with the conspiracy blindness hypothesis that people make errors when they do not take the time to think about their decisions, and inconsistent with the

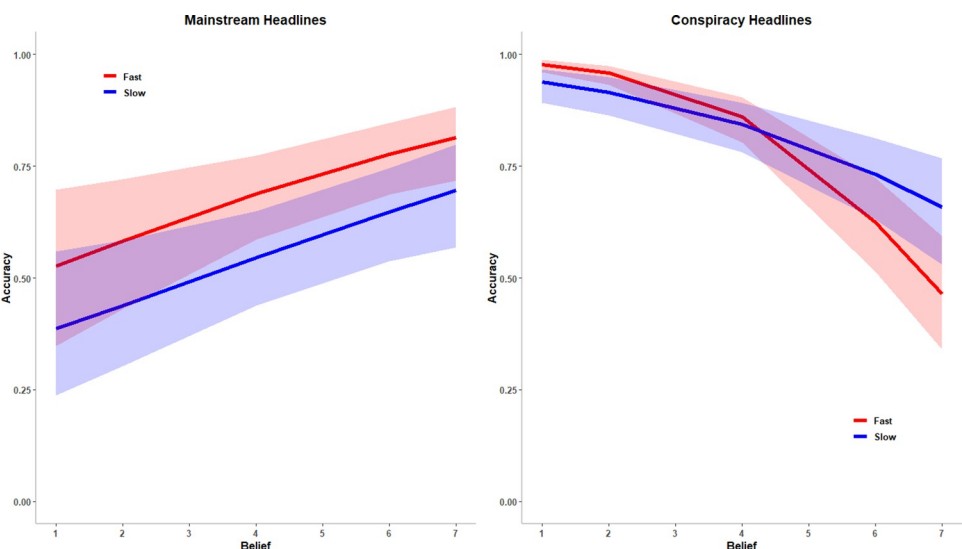

**Fig 1. The three-way interaction between belief, article type, and response time predicting accuracy.** *Note.* Response time is graphed as 1SD below the mean (labelled "fast") and 1 SD above the mean (labelled "slow"). The error bars represent the 95% confidence interval. The x-axis represents the original 1 (Completely false) to 7 (Completely true) scale for belief before rescaling. Figures created using lme4 version 1.1–23 [47] and ggplot2 version 3.30 [51].

motivated reasoning prediction that people make errors when they take the time to engage in motivated reasoning.

As predicted by both the motivated reasoning and conspiracy blindness hypotheses, there was a three-way interaction between belief, article type, and response time predicting accuracy (Fig 1). There was no interaction between belief and response time for mainstream headlines, β = 0.01, *SE* = 0.05, *p* = .883, *OR* = 1.01, Cohen's *d* = 0.005. In other words, the more participants believed the mainstream articles, the more accurate they were in recognizing that they did not contain conspiracy theories, regardless of how long they took to come to that conclusion. There was, however, an interaction between belief and response time for conspiracy articles (β = 0.13, *SE* = 0.04, *p* = .003, *OR* = 1.14, Cohen's *d* = 0.07) such that the more participants believed a conspiracy article and the less time they took to decide if it contained a conspiracy theory, the less likely they were to recognize that it contained a conspiracy theory, β = -0.98, *SE* = 0.09, *p* < .001, *OR* = 0.37, Cohen's *d* = -0.55. When participants believed a conspiracy article and took more time to decide if it contained a conspiracy theory, the relationship between their belief and their ability to recognize that the article contained a conspiracy theory was weaker, β = -0.51, *SE* = 0.10, *p* < .001, *OR* = 0.60, Cohen's *d* = -0.28. These results suggest that people have difficulty recognizing that their beliefs are conspiracy theories when they do not take the time to consider if they might be conspiracy theories. When they take more time to consider if their beliefs might be conspiracy theories, they are more likely to recognize when they are conspiracy theories. This pattern of results is consistent with the conspiracy blindness hypothesis that people have difficulty recognizing that their beliefs are conspiracy theories because they do not take the time to consider whether those beliefs might be conspiracy theories.

## Discussion

Study 1 supported the motivated reasoning and conspiracy blindness hypotheses' predictions that people who believe conspiracy theories have difficulty recognizing their beliefs are

conspiracy theories. The more participants believed the conspiracy articles, the more difficulty they had recognizing that they contained conspiracy theories, but only when they did not take much time to consider their answer. People were least likely to decide that conspiracy theories are indeed conspiracy theories when they (1) believed them and (2) did not take much time to think about their decision. This supports the conspiracy blindness hypothesis that people cannot recognize the conspiracy theories they believe as conspiracy theories because they do not bother to consider whether their beliefs could be conspiracy theories.

Inconsistent with my predictions for both hypotheses, I found that speed predicted accuracy for mainstream articles. Participants were better at recognizing that mainstream articles did not contain conspiracy theories when they took less time to decide and worse when they took more time.

Although the results of Study 1 were most consistent with the conspiracy blindness hypothesis, it sacrificed internal validity for environmental validity. As Study 1 used real headlines, it was not always obvious if they contained conspiracy theories if the participants skipped the summaries. Study 1 also expected participants to use their own definition for what constitutes a "conspiracy theory" to determine if the articles contained conspiracy theories. Although this reflects how people evaluate information in their day-to-day lives, it may be that participants were "less accurate" overall because their personal definition of "conspiracy theory" differed from the definition commonly used in academia. I conducted a second study to address these concerns and to provide a second test of the conspiracy blindness hypotheses.

## Study 2

Study 1 provided an initial test of the motivated reasoning and conspiracy blindness hypotheses and found more support for the conspiracy blindness hypothesis- that people have difficulty recognizing the conspiracy theories they believe as conspiracy theories because they do not take time to consider it. However, some of its findings may have been artefacts of its stimuli. It may have been difficult to identify the conspiracy theories in the article summaries. The conspiracy theories in the articles in Study 1 were also somewhat limited in scope because all the headlines were collected from the same six-month time period. To address this, Study 2 used a variety of manufactured statements instead of article summaries to test if the findings from Study 1 would replicate with more internally valid stimuli. These statements were intentionally written to contain (or not contain) overt conspiracy theories about a variety of topics from a variety of time periods. As with Study 1, these conspiracy theories were not necessarily false. They fit the definition of "conspiracy theory" used in this paper, and ran counter to the mainstream narrative at the time this study was conducted.

Study 1 also assumed that all participants had a definition of "conspiracy theory" that was consistent with the one typically used in the academic literature and in this paper. Discrepancies between participants' definitions may explain some of the results in Study 1. For example, if participants think that conspiracy theories are necessarily false, then they would not recognize conspiracy theories that they believed were true as conspiracy theories. Study 2 manipulated whether participants were provided with the literature's definition for conspiracy theories or were not provided with a definition, to see if access to a standardized definition would facilitate recognition.

This manipulation also allowed for a stronger test of the conspiracy blindness hypothesis. If the conspiracy blindness hypothesis is true, then participants should make more accurate decisions when they are asked to apply the definition for conspiracy theory to the statements they read, because this task forces them to consider the possibility that their belief is a conspiracy theory when they would not have considered it otherwise.

## Method

My hypotheses suggested a significant four-way interaction between statement type (No Conspiracy or Conspiracy), belief, response time, and definition condition (No Definition or Definition). I conducted a power analysis for this four-way interaction. The power analysis concluded that a sample of 250 would be sufficient for more than 80% power with an alpha of .05, based on a Monte Carlo simulation (N = 200 samples) with an estimated effect size of Cohen's $d$ = .50 using the simr package (version 1.0.5 [40]). This estimated effect size is based on the results of two similar studies that examined the effect of information processing (i.e., how critically participants thought about their answers) and ideological congeniality (i.e., how consistent the information was with participants' pre-existing beliefs) on participants' ability to identify fake news [38]. This sample size was decided upon before data collection began, and data collection ended before analysis began.

### Participants

Two hundred and fifty-one participants from Amazon Mechanical Turk (mTurk) completed the survey between September 16[th] and September 17[th], 2020. The average age of the sample was 35.48 ($SD$ = 11.37). One-hundred and sixty-four participants identified as men, 85 identified as women, and two identified as nonbinary. The majority of the sample ($n$ = 141) had a four-year college degree. Fifty-two had a professional degree, 29 had some college, 12 had a two-year degree, 14 were high school graduates, two had a doctorate, and one had less than a high school degree. One-hundred and sixty-seven participants identified as White, 47 identified as Black or African American, 19 identified as Asian, 7 identified as Hispanic or Latino, one identified as something else, one identified as American Indian or Alaska Native, and eight identified as multiracial. On a scale of -3 (Very Liberal) to 3 (Very Conservative) the sample averaged 0.26 ($SD$ = 1.96). Ninety participants identified on the liberal side of the spectrum, 122 identified on the conservative side of the spectrum, and 39 identified as moderate.

All participants provided informed consent via an electronic consent form. This research was approved by the University of Illinois at Chicago's Institutional Review Board under protocol #2020–1103.

### Procedure

Upon beginning the survey, participants were randomly assigned to one of two conditions: The conspiracy definition condition and the no definition condition. Participants in the conspiracy definition condition were asked to read the definition of "conspiracy theory" used in the introduction of this paper. In the no definition condition, participants did not read a definition for "conspiracy theory."

After reading (or not reading) the definition, participants read instructions for the rest of the study accompanied by an example statement and response to ensure that they understood the task. After they read the instructions, they were presented with twenty randomized statements selected from the thirty tested in the pilot study (S1–S4 Text). Ten of these statements were conspiracy theories, and ten of these statements were not conspiracy theories. All statements are available in S2 Table, and selection criteria are described in the pilot study in the supplement. Participants viewed the statements in a random order, and each statement was presented on its own page of the survey. In the conspiracy definition condition, the definition for "conspiracy theory" was available above each statement. In the no definition condition, there was no definition above the statement.

To ensure that participants applied the provided definition to the statements, participants completed a checklist for each statement. This checklist contained the three major features of

**Table 4. Means, standard deviations, and correlations with confidence intervals for key variables.**

| Variable | M | SD | 1 | 2 | 3 | 4 | 5 | 6 |
|---|---|---|---|---|---|---|---|---|
| 1. Age | 35.48 | 11.37 | | | | | | |
| 2. Education | 4.76 | 1.12 | .01 | | | | | |
| | | | [-.11, .14] | | | | | |
| 3. Political Orientation | 0.26 | 1.96 | .09 | .08 | | | | |
| | | | [-.03, .22] | [-.05, .20] | | | | |
| 4. Mainstream Belief | 1.24 | 0.97 | -.10 | -.18** | -.15* | | | |
| | | | [-.22, .02] | [-.29, -.05] | [-.27, -.03] | | | |
| 5. Conspiracy Belief | 0.34 | 1.16 | -.25** | .04 | .26** | .03 | | |
| | | | [-.36, -.13] | [-.08, .17] | [.14, .37] | [-.09, .15] | | |
| 6. Reaction Time | 4.79 | 5.86 | .04 | .09 | .02 | -.09 | .11 | |
| | | | [-.08, .17] | [-.04, .21] | [-.10, .15] | [-.21, .04] | [-.01, .23] | |
| 7. Score | 13.76 | 4.12 | .08 | -.25** | -.27** | .52** | -.50** | -.14* |
| | | | [-.04, .21] | [-.37, -.14] | [-.38, -.15] | [.43, .61] | [-.59, -.40] | [-.26, -.02] |

*Note*. Values in square brackets indicate the 95% confidence interval for each correlation coefficient.

* indicates $p < .05$.

** indicates $p < .01$. "Mainstream Belief" is the average reported belief for all "no conspiracy" items. "Conspiracy belief" is the average reported belief for all conspiracy items. Score represents the total number of statements participants correctly classified as "no conspiracy" or "conspiracy."

the definition for "conspiracy theory": 1) a group of powerful people is working together to accomplish a goal, 2) while attempting to keep their activity secret, and 3) acting at the expense of others. Participants indicated every feature they believed was present in that statement. If they did not believe that any of the features were present in the statement, they had the option to select a fourth box that read "none of the three apply." All participants completed this checklist, regardless of their condition. Participants in the no definition condition were not told what the checklist represented.

Once they completed the checklist, participants were asked whether the statement was a conspiracy theory (yes or no). The survey software recorded how long it took them to make their decision. Finally, to measure how much participants believed each statement, participants rated each statement on a scale of 1 (Completely False) to 7 (Completely True). After responding to all questions for all twenty statements, participants provided demographic information including their age, gender, race and ethnicity, political orientation, and education. As in Study 1, participants' belief in each statement was re-scored to a scale of -3 (Completely false) to 3 (Completely true) and centered on 0 such that negative scores indicated disbelief and positive scores indicated belief. Participants' political orientation was rescored to a scale of -3 (Very liberal) to 3 (Very conservative) and centered on 0 such that negative scores indicated a liberal political orientation, and positive scores indicated a conservative political orientation. The full survey is available in the supplemental material.

## Results

The means and correlations between all key and demographic variables are available in Table 4. Some of the correlations between variables in Study 2 were different than in Study 1. Whereas education was positively correlated with belief in conspiracy article summaries in Study 1, it was not correlated with belief in conspiracy statements in Study 2. Although belief in mainstream article summaries was positively correlated with belief in conspiracy article summaries in Study 1, belief in mainstream statements was not correlated with belief in

**Table 5. Multilevel model results predicting the likelihood of a correct answer from belief, article type, response time, and condition.**

| | Model 1 | Model 2 | Model 3 | Model 4 | Model 5 | Model 6 | Model 7 | Model 8 |
|---|---|---|---|---|---|---|---|---|
| (Intercept) | 1.68 (0.15) *** | 1.67 (0.15) *** | 1.51 (0.13) *** | 1.51 (0.13) *** | 1.51 (0.13) *** | 1.51 (0.13) *** | 1.49 (0.13) *** | 1.50 (0.13) *** |
| Belief | -0.12 (0.06)* | -0.09 (0.06) | -0.01 (0.06) | -0.02 (0.06) | -0.02 (0.06) | -0.02 (0.06) | -0.00 (0.06) | -0.00 (0.06) |
| Statement Type | | 1.77 (0.36) *** | 1.79 (0.35) *** | 1.79 (0.35) *** | 1.80 (0.35) *** | 1.77 (0.35) *** | 1.77 (0.35) *** | 1.72 (0.35) *** |
| Belief * Statement Type | | | -0.94 (0.13) *** | -0.93 (0.13) *** | -0.93 (0.13) *** | -0.93 (0.13) *** | -0.88 (0.13) *** | -0.87 (0.13) *** |
| Response time | | | | -0.09 (0.05)` | -0.09 (0.05)` | -0.08 (0.05)` | -0.09 (0.05)` | -0.08 (0.05) |
| Belief * Response time | | | | | 0.06 (0.05) | 0.06 (0.05) | 0.03 (0.06) | 0.03 (0.06) |
| Statement Type * Response time | | | | | 0.26 (0.10)* | 0.25 (0.10)* | 0.24 (0.10)* | 0.24 (0.10)* |
| Belief * Statement Type * Response time | | | | | 0.19 (0.11)` | 0.19 (0.11)` | 0.18 (0.11) | 0.18 (0.11) |
| Condition | | | | | | -0.08 (0.22) | -0.08 (0.22) | 0.00 (0.21) |
| Belief * Condition | | | | | | | 0.15 (0.13) | 0.13 (0.13) |
| Statement Type * Condition | | | | | | | 1.57 (0.65)* | 1.53 (0.67)* |
| Response time * Condition | | | | | | | 0.08 (0.10) | 0.06 (0.10) |
| Belief * Statement Type * Condition | | | | | | | 0.22 (0.26) | 0.24 (0.26) |
| Belief * Response time * Condition | | | | | | | -0.08 (0.11) | -0.07 (0.11) |
| Statement Type * Response time * Condition | | | | | | | -0.03 (0.21) | -0.02 (0.21) |
| Belief * Statement Type * Response time * Condition | | | | | | | -0.53 (0.22)* | -0.56 (0.23)* |
| Education | | | | | | | | -0.56 (0.11) *** |
| Political Orientation | | | | | | | | -0.39 (0.10) *** |
| AIC | 4182.44 | 4160.88 | 4109.73 | 4108.35 | 4103.88 | 4105.64 | 4104.20 | 4068.56 |
| BIC | 4215.05 | 4200.01 | 4155.38 | 4160.52 | 4175.61 | 4183.89 | 4228.11 | 4205.51 |
| Log Likelihood | -2086.22 | -2074.44 | -2047.87 | -2046.18 | -2040.94 | -2040.82 | -2033.10 | -2013.28 |
| **Random Effects** | | | | | | | | |
| Intercept | 2.63 | 2.60 | 2.02 | 1.98 | 1.98 | 1.97 | 1.98 | 1.56 |
| Statement Type | 25.00 | 23.18 | 22.83 | 22.83 | 22.19 | 22.82 | 21.91 | 22.89 |
| Statement | 0.11 | 0.08 | 0.06 | 0.06 | 0.06 | 0.06 | 0.06 | 0.06 |

*Note.* * indicates a *p*-value that is less than .05

** indicates a *p*-value that is less than .01, and

*** indicates a *p*-value that is less than .001. Standard errors are presented in parentheses. *p*-values are based on asymptotic Wald tests (lme4 ver. 1.1–23; Bates et al., 2015). This analysis did not exclude responses with abnormally long response times. The maximum response time for any item for any participant was 1013.90 seconds. A later analysis that excluded items with response times three standard deviations above the mean did not change the overall pattern of results and the four-way interaction remained significant, *p* = .027.

conspiracy statements in Study 2. The means and standard deviations of belief for individual statements, and correlations with all demographic and key variables are available in the supplemental material (S5 Table). Histograms of response time by statement type and definition, as well as violin plots of response time for each statement, are also available in the supplemental material (S1–S5 Figs).

The conspiracy blindness hypothesis predicted a four-way interaction between belief, statement type, response time, and definition condition. Additionally, the conspiracy blindness hypothesis predicted that there would not be an interaction between belief and reaction time for mainstream article summaries, but that there would be an interaction between belief and

**Table 6. Summary of comparisons between models, testing whether adding terms improved model fit.**

| Model Comparison | Change in *df* | Deviance | $\chi^2$ | *p* |
|---|---|---|---|---|
| *Model 1 versus Model 0 (Null)*<br>Belief | +1 | 4172.44 | 3.88 | .049 |
| *Model 2 versus Model 1*<br>Belief + Article Type | +1 | 4148.88 | 23.56 | < .001 |
| *Model 3 versus Model 2*<br>Belief * Article Type | +1 | 4095.73 | 53.15 | < .001 |
| *Model 4 versus Model 3*<br>Belief * Article Type + Response Time | +1 | 4092.35 | 3.38 | .066 |
| *Model 5 versus Model 3*<br>Belief * Article Type * Response Time | +4 | 4081.88 | 13.85 | .008 |
| *Model 6 versus Model 5*<br>Belief * Article Type * Response Time + Condition | +1 | 4081.64 | 0.24 | .621 |
| *Model 7 versus Model 5*<br>Belief * Article Type * Response Time * Condition | +8 | 4066.20 | 15.67 | .047 |
| *Model 7 versus Model 8*<br>Belief * Article Type * Response Time * Condition + Education + Political Orientation | +1 | 4040.41 | 25.79 | < .001 |

reaction time for conspiracy article summaries. The conspiracy blindness hypothesis predicted that participants would have difficulty recognizing the conspiracy theories they believed as conspiracy theories when they made fast decisions and did not have access to the definition, but the interaction between belief and response time would disappear when participants had access to the definition.

To test the conspiracy blindness hypothesis, I used a logistic mixed-effects model. I took a forward-fitting approach, adding the predictors in stages [46]. I performed all analyses using the lme4 package in R (lme4 ver. 1.1–23 [47]). All models used the same random effects structure. The random effects were crossed. Statement (item) was nested within statement type (no conspiracy or conspiracy), and statement type was nested within subjects. Statement was treated as a repeated measure for subjects, but a nested variable for statement type [48,49]. Statements and subjects were allowed their own intercepts, and statement type was entered as a random slope. I blocked the correlation between subjects and statement type. Model 8, which included the main effects of belief, statement type, response time, condition, and their four-way interaction as well as control variables, was the best fit. All model coefficients are available in Table 5.

Model 0 contained no fixed effects. Model 1 tested for a relationship between belief (mean centered and *z*-scored) and accuracy (whether participants correctly categorized the statement as a conspiracy article or a mainstream article). Model 2 included statement type (no conspiracy, coded as -0.5 or conspiracy, coded as 0.5). Model 3 added the interaction between belief and statement type. Model 4 added response time (log 10 transformed, mean centered, and *z*-scored). Model 5 added the three-way interaction between belief, statement type, and response time. Model 6 added the fixed effect of definition condition (no definition, coded as -0.5, or conspiracy definition, coded as 0.5). Model 7 added the four-way interaction between belief, statement type, response time, and definition condition. Model 8 added education (*z*-scored) and political orientation (centered on the scale's midpoint and standardized) as level two control variables. All model comparisons are available in Table 6.

Only one of the four main effects, statement type, was significant. Participants were more accurate when deciding if conspiracy statements contained conspiracy theories than if mainstream statements contained conspiracy theories. There were no main effects of belief, response time, or condition.

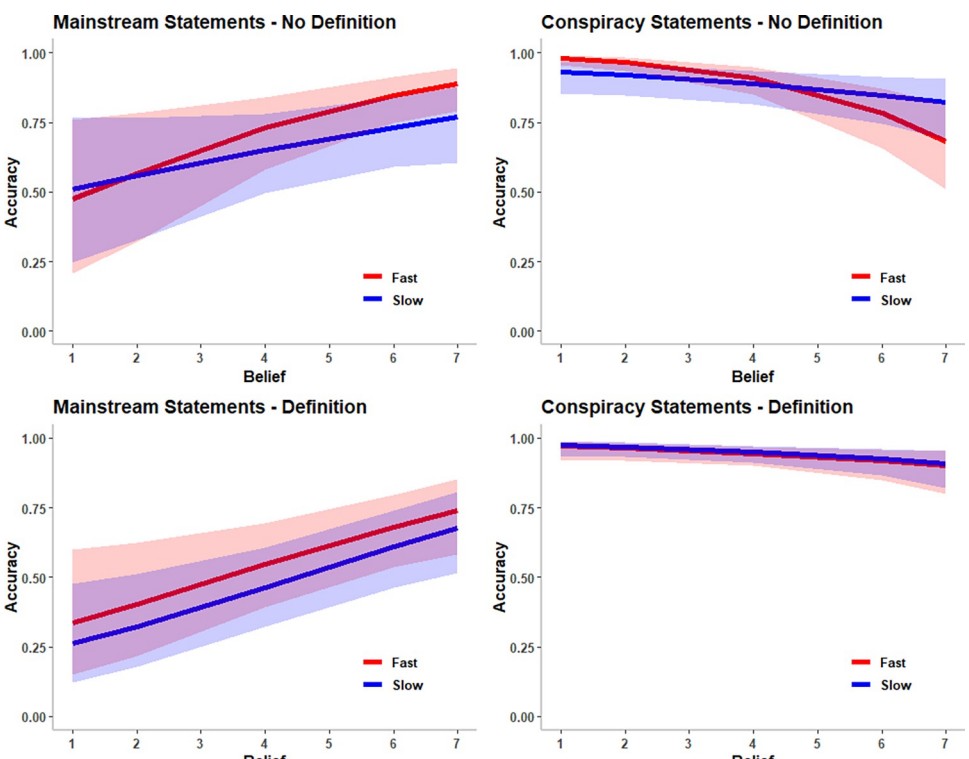

**Fig 2. The four-way interaction between belief, statement type, response time, and condition predicting accuracy.**
*Note*. Response time is graphed as 1SD below the mean (labelled "fast") and 1 SD above the mean (labelled "slow").
The error bars represent the 95% confidence interval The x-axis represents the original 1 (Completely false) to 7
(Completely true) scale for belief before rescaling. Figures created using lme4 version 1.1–23 [47] and ggplot2 version
3.30 [51].

There was a two-way interaction between belief and article type that replicated the interaction in Study 1. Consistent with the conspiracy blindness hypotheses, the more participants believed the conspiracy statements, the less likely they were to recognize that they contained conspiracy theories ($\beta$ = -0.43, *SE* = 0.08, *p* < .001, *OR* = 0.65, Cohen's *d* = -0.24), and the more participants believed the mainstream statements, the more likely they were to correctly recognize that they did not contain conspiracy theories, $\beta$ = 0.43, *SE* = 0.10, *p* < .001, *OR* = 1.54, Cohen's *d* = 0.24 (Simple slopes calculated using emmeans ver 1.5.3 [52]). The interaction between statement type and response time also replicated the interaction observed in Study 1. The amount of time participants took to decide whether a statement contained a conspiracy theory did not predict the accuracy of their decision for conspiracy statements ($\beta$ = 0.04, *SE* = 0.07, *p* = .607, *OR* = 1.04, Cohen's *d* = 0.02), but faster decisions predicted more accurate decisions for mainstream statements, $\beta$ = -0.20, *SE* = 0.07, *p* = .006, *OR* = 0.82, Cohen's *d* = -0.11. This result is inconsistent with the conspiracy blindness hypotheses that there would no relationship between response time and accuracy for mainstream headlines and a relationship (negative for the motivated reasoning hypothesis or positive for the conspiracy blindness hypothesis) between response time and accuracy for conspiracy headlines. There were no other two-way interactions. Unlike in Study 1, none of the three-way interactions were significant.

Finally, there was a four-way interaction between belief, statement type, response time, and definition condition (Fig 2). There was no interaction between belief and response time for mainstream statements for participants who did not receive the definition ($\beta$ = -0.14,

*SE* = 0.08, *p* = .105, *OR* = 0.87, Cohen's *d* = -0.08), for mainstream statements for participants who did receive the definition (β = 0.01, *SE* = 0.07, *p* = .827, *OR* = 1.01, Cohen's *d* = 0.005), or for conspiracy statements for participants who did receive the definition, β = -0.03, *SE* = 0.08, *p* = .683, *OR* = 0.97, Cohen's *d* = -0.02. In other words, how long participants took to decide whether a statement contained a conspiracy theory did not affect the relationship between belief and accuracy for mainstream statements at all, or for conspiracy statements when participants had the definition for "conspiracy theory."

There was, however, an interaction between belief and response time for conspiracy statements for participants who did not receive the definition, β = 0.23, *SE* = 0.07, *p* = .001, *OR* = 1.26, Cohen's *d* = 0.13. The interaction replicated the interaction between belief and response time for conspiracy articles in Study 1. The more participants believed a conspiracy statement and the less time they took to decide if it contained a conspiracy theory, the less likely they were to recognize that the statement contained a conspiracy theory, β = -0.85, *SE* = 0.14, *p* < .001, *OR* = 0.43, Cohen's *d* = -0.46. Belief was not related to accuracy when they took more time to think about their decision, β = -0.26, *SE* = 0.15, *p* = .078, *OR* = 0.77, Cohen's *d* = -0.14.

In short, the results replicated Study 1 and mostly supported the conspiracy blindness hypothesis. Participants had difficulty identifying the conspiracy theories they believed as conspiracy theories when they did not take the time to consider whether they might be conspiracy theories. However, when participants were forced to consider whether their beliefs might be conspiracy theories (by actively applying the definition for "conspiracy theory" to the statement), or when they took more time to think, they were able to recognize them as conspiracy theories. Replicating prior research, access to the definition for belief in conspiracy theories did not reduce belief in conspiracy theories, *t*(249) = 1.57, *p* = .118.

## Discussion

Study 2 replicated the results of Study 1. Once again, the results were most consistent with the conspiracy blindness hypothesis: The more participants believed the conspiracy statements, the less likely they were to recognize that they contained conspiracy theories, but only when they did not take much time to consider their answer. People were least likely to correctly identify conspiracy statements when they did not think very long (or perhaps at all) about whether the statement contained a conspiracy theory.

Study 2 provided additional support for the conspiracy blindness hypothesis by demonstrating that, when people take the time to consider whether their beliefs are conspiracy theories, they can recognize them as conspiracy theories. In other words, it seems that participants' difficulty recognizing the conspiracy theories they believed as conspiracy theories was due to a lack of consideration rather than motivated reasoning. Although people may not realize that some of their beliefs are conspiracy theories, if they critically examine their beliefs and are given the right tools, they can recognize them for what they are.

## General discussion

This collection of studies tested whether people can recognize their own conspiracy beliefs as conspiracy theories using headlines from real online articles (Study 1) and tailored statements (Study 2). Both Studies 1 and 2 demonstrated that people have difficulty recognizing that the conspiracy theories that they believe are, in fact, conspiracy theories. In addition to demonstrating this bias, Studies 1 and 2 examined its source. I tested two possible explanations for the failure to recognize that one believes a conspiracy theory: 1) The *motivated reasoning hypothesis*, that people do not recognize their own beliefs as conspiracy theories because they

take the time to rationalize why their belief is not an "irrational" conspiracy theory, and 2) the *conspiracy blindness hypothesis* that people do not recognize their own beliefs as conspiracy theories because they do not take the time to consider whether their belief might be a conspiracy theory.

The results of both Studies 1 and 2 were more consistent with the conspiracy blindness hypothesis than the motivated reasoning hypothesis. Participants who took less time to decide if their beliefs were conspiracy theories were more likely to incorrectly decide that their conspiracy beliefs were not conspiracy theories (consistent with the conspiracy blindness hypothesis) and participants who took more time to decide if their beliefs were conspiracy theories were less likely to incorrectly decide that their conspiracy beliefs were not conspiracy theories (contrary to the motivated reasoning hypothesis). Study 2 provided further support for the conspiracy blindness hypothesis by demonstrating that when people were forced to stop and consider whether their beliefs might be conspiracy theories, their conspiracy blindness was significantly reduced. Specifically, when participants were forced to evaluate their conspiracy beliefs using a check list of features of conspiracy theories and a definition for "conspiracy theory," they were able to correctly identify conspiracy theory features over 90% of the time.

The ability to overcome conspiracy blindness, however, does not guarantee a reduction in conspiracy belief. In Study 2, participants reported their belief in each statement after reporting whether they believed the statement contained a conspiracy theory. Many participants were still willing to report that they thought the conspiracy statements were "completely true," even when they acknowledged that they contained conspiracy theories. This is consistent with previous findings that labeling peoples' conspiracy beliefs as conspiracy theories does not reduce self-reported belief in those conspiracy theories, and inconsistent with the motivated reasoning hypothesis that people are attempting to distance themselves from stigmatized beliefs.

Although overcoming conspiracy blindness may not change belief, it may change behavior. Calling a claim a conspiracy theory in a public setting (e.g. one member of British Parliament calling another member's claim a conspiracy theory) can reduce repetitions of that claim in that setting [12]. Overcoming conspiracy blindness may not reduce belief in conspiracy theories, but if people are aware that some of their beliefs are conspiracy theories, they may be less likely to share those beliefs, thereby slowing the spread of conspiracy theories, for better or for worse. That said, McKenzie-McHarg et al. [12]'s findings come from a very specific context (British parliamentary debates). Further research should examine whether overcoming conspiracy blindness can deter people from repeating a conspiracy theory in other contexts, such as on social media or during informal social gatherings.

Although people can recognize that others would consider their beliefs conspiracy theories, they may not agree with them. Consistent with this possibility, one participant in Study 2 mentioned that they were not sure "whether [they were] supposed to be using [their] own judgment" or whether they were supposed to use the study's definition for "conspiracy theory" to make their decision about whether a given belief was a conspiracy theory. Other, more recent research demonstrates that many peoples' personal definitions for "conspiracy theory" may prevent them from concluding that anything they believe to be true is a conspiracy theory (e.g., they believe that a conspiracy theory is always false [53]). This may be one factor that contributed to the negative relationship between conspiracy belief and ability to recognize conspiracy theories. However, if the discrepancy between the academic definition and colloquial definitions for "conspiracy theory" were the only factor, the amount of time participants took to make their decisions should be unrelated to their accuracy. So, while participants' personal definitions may have contributed to the relationship between belief and accuracy, how much they considered their answers was also a contributing factor.

While having access to a unified definition for "conspiracy theory" increased participants' ability to recognize conspiracy theories regardless of their belief, providing people with a unified definition may have negative consequences. Having access to the definition for "conspiracy theory" in Study 2 made participants more likely to label mainstream information as a conspiracy theory. This finding raises concerns about attempts to decrease conspiracy blindness, debunk false conspiracy theories, and research that tests methods of debunking false conspiracy theories (e.g., [54,55]). Few studies in the debunking literature examine how efforts to debunk misinformation influences perceptions of true information (e.g., [56,57]). Although debunking strategies may reduce belief in false conspiracy theories, they may also reduce belief in mainstream information from official sources and true conspiracy theories (for similar findings with misinformation more generally, see [58]).

In summary, the results of the studies presented here suggest that people experience conspiracy blindness: They have difficulty recognizing their own beliefs as conspiracy theories. Conspiracy theory blindness is not due to a conscious aversion to labeling beliefs "conspiracy theories," but exists because people are not self-critical of their beliefs. Although they recognize that conspiracy theories and theorists exist, it does not occur to them that they might be one of them.

## Supporting information

**S1 Fig. Histograms of response time in seconds by article summary type.**
(TIF)

**S2 Fig. Violin plots of response time by article summary.**
(TIF)

**S3 Fig. Odds ratios for Model 7.**
(TIF)

**S4 Fig. Histograms of response time in seconds by statement type and definition condition.**
(TIF)

**S5 Fig. Violin plots of response time by statement.**
(TIF)

**S6 Fig. Odds ratios for Model 9.**
(TIF)

**S1 Table. Means and standard deviations of belief, the proportion of participants who saw all three conspiracy features, correlations between political orientation and belief, and interrater agreement on the presence of all three conspiracy features.**
(DOCX)

**S2 Table. Means and standard deviations of belief, the proportion of participants who saw all three conspiracy features, correlations between political orientation and belief, and interrater agreement on the presence of all three conspiracy features.**
(DOCX)

**S3 Table. Means and standard deviations for belief in each article summary and correlations with demographic variables in Study 1.**
(DOCX)

**S4 Table. A full list of headlines for the article summaries in Study 1.**
(DOCX)

**S5 Table. Means and standard deviations for belief in each statement and correlations with demographic variables in Study 2.**
(DOCX)

**S6 Table. A full list of statements in Study 2.**
(DOCX)

**S1 Text. Pilot study.** A description of the pilot study used to select stimuli.
(DOCX)

**S2 Text. Pilot study survey.** The full text of the pilot study survey.
(DOCX)

**S3 Text. Study 1 survey.** The full text of the Study 1 survey.
(DOCX)

**S4 Text. Study 2 survey.** The full text of the Study 2 survey.
(DOCX)

## Acknowledgments

### Open practices

The experiments in this article are eligible for Open Materials and Open Data badges. All data and materials are available on the Open Science Framework at the following link: https://osf.io/h3gdr/?view_only=c092855b387243269fc3ac9751b4fff9.

## Author Contributions

**Conceptualization:** J. P. Prims.

**Data curation:** J. P. Prims.

**Formal analysis:** J. P. Prims.

**Investigation:** J. P. Prims.

**Methodology:** J. P. Prims.

**Project administration:** J. P. Prims.

**Resources:** J. P. Prims.

**Software:** J. P. Prims.

**Validation:** J. P. Prims.

**Visualization:** J. P. Prims.

**Writing – original draft:** J. P. Prims.

**Writing – review & editing:** J. P. Prims.

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
