## [Decision Letter · Decision Letter 0]

30 Jan 2024

PONE-D-23-41311Call it a Conspiracy: How Conspiracy Belief Predicts Recognition of Conspiracy TheoriesPLOS ONE

Dear Dr. Prims,

Thank you for submitting your manuscript to PLOS ONE. After careful consideration, we feel that it has merit but does not fully meet PLOS ONE’s publication criteria as it currently stands. Therefore, we invite you to submit a revised version of the manuscript that addresses the points raised during the review process.

We look forward to receiving your revised manuscript.

Kind regards,

Prof. Anat Gesser-Edelsburg, Ph.D.

Academic Editor

PLOS ONE

2. Please include a copy of Table 3 which you refer to in your text on page 16.

3. Please upload a copy of Supplementary tables which you refer to in your text on pages 10 and 25.

Comments from PLOS Editorial Office: We note that one or more reviewers has recommended that you cite specific previously published works. As always, we recommend that you please review and evaluate the requested works to determine whether they are relevant and should be cited. It is not a requirement to cite these works. We appreciate your attention to this request.

Reviewers' comments:

Reviewer's Responses to Questions

**Comments to the Author**

1. Is the manuscript technically sound, and do the data support the conclusions?

Reviewer #1: Yes

Reviewer #2: Partly

2. Has the statistical analysis been performed appropriately and rigorously? 

Reviewer #1: Yes

Reviewer #2: Yes

3. Have the authors made all data underlying the findings in their manuscript fully available?

Reviewer #1: Yes

Reviewer #2: Yes

4. Is the manuscript presented in an intelligible fashion and written in standard English?

Reviewer #1: Yes

Reviewer #2: Yes

5. Review Comments to the Author

Reviewer #1: This is an excellent study. A few minor comments:

1. I think the point could be brought out more that people believe propositions because they think those propositions are true. Since the term conspiracy theory often implies to people that the idea is not true, rarely will people label their own ideas as conspiracy theories. Usually, people like to label other people's ideas a conspiracy theories.

2. Some of the citations were a bit dated and there is some literature that might help make your case. You should cite:

Thalmann, Katharina. 2019. The Stigmatization of Conspiracy Theory since the 1950s:" A Plot to Make Us Look Foolish". New York: Routledge.

Douglas, Karen, Jan-Willem van Prooijen, and Robbie M. Sutton. 2022. "Is the Label ‘Conspiracy Theory’ a Cause or a Consequence of Disbelief in Alternative Narratives?" British Journal of Psychology n/a.

Dentith, M. R. X., G. Husting, and M. Orr. 2023. "Does the Phrase “Conspiracy Theory” Matter?" Society.

3. The writingon the front end could be tightened.

The Heaton quote can go.

Most people believe one or a few conspiracy theories; Oliver and Wood only got 55% because they only asked about 7 cts. If you ask about more cts on a survey, more people believe at least one.

Cts are associated with harms; avoid making the causal argument: Uscinski, Joseph, Adam M. Enders, Casey Klofstad, and Justin Stoler. 2022. "Cause and Effect: On the Antecedents and Consequences of Conspiracy Theory Beliefs." Current Opinion in Psychology 101364. You could just cite the Jolley, Douglas, and Mari piece for consequences too.

4. The normative point should be made in the final discussion section: If an idea qualifies as a a conspiracy theory, does that mean that belief in it is unwarranted? The literature suggests yes Keeley 1999; Levy 2207; Uscinski and Enders 2023. At the same time, being a conspriacy theory does not mean false; rather it just means not proven in some way.

Reviewer #2: This is an interesting paper but I have some worries about the conceptualisation of both the theoretical framework the paper relies upon, and both of the studies.

Let me start with the theoretical framework. The paper assumes that what philosopher Charles Pigden calls a "modern superstition" about conspiracy theories is right - these are theories which are irrational. This is a bit of a problem for the paper, for two reasons: 1. The paper works with a fairly non-pejorative or evaluatively-laden definition of conspiracy theory ("conspiracy theories are explanations for events or circumstances that claim a group of powerful people is working together to accomplish a goal that comes at the expense of others, while attempting to keep their actions or intentions a secret") which admits in true conspiracies as being the subject of conspiracy theories; thus some of the presentation of material in the paper is at odds with itself when it assumes conspiracy theories are bad despite working with a definition that takes no sides. 2. The paper claims "These theories, like the people who believe them, are often considered irrational (McKenzie-McHarg & Fredheim, 2017; Wood & Douglas, 2013), even by the people who study them (Basham & Dentith, 2016; Bjerg & Presskorn-Thygesen, 2017)" - this is a problem because this is a misreading of Basham and Dentith, and probably not a kind reading of Bjerg & Presskorn-Thygesen. These authors have, to varying degrees, argued that the idea conspiracy theories and conspiracy theorists are irrational doesn't hold water when you examine said theories or said theorists.

The paper, as it stands, uses the "often" claim in a way that is only representative depending on what literature you admit or do not admit into the pool: in philosophy and sociology, for example, it is not considered prima facie irrational. I would recommend conditionalising this claim to say that in psychology and some of the social sciences the theories and the people who believe them are considered to be irrational, although in other fields of study around conspiracy theories, this claim is hotly contested. See the most recent special issue of Social Epistemology on this topic for details: https://www.tandfonline.com/toc/tsep20/37/4

This kind of thing is important, especially given the evident misreading of at least one article cited in this paper. It is laudable that the paper works with a non-pejorative definition of conspiracy theory, and so it is important that if the paper wants to pursue an analysis of conspiracy theories under the lens that they are irrational, then the paper needs to point out that despite the value neutral definition some scholars who study conspiracy theories treat them as prima facie or largely irrational. The "some" here is important, as the irrationality of conspiracy theory belief generally is, it turns out, contested.

I also want to push back on another aspect of the pejorative tilt this paper takes: we are told "Belief in conspiracy theories can have serious consequences for public health, the environment, and democracy" and given a list of negative social consequences. But even scholars like Karen Douglas and Robbie Sutton have admitted that there are positive social consequences to belief in conspiracy theories: such as detecting conspiracies (on the definition proposed by this paper Woodward and Bernstein believed a conspiracy theory about the Watergate Affair, one that turned out to be true). Once again, the paper seems to be accepting at face value what merely some of the literature says about conspiracy theories.

This brings me to the short section on the motivating reason hypothesis, and it's slight discussion on labelling practices: there is already quite a lot of literature on the label "conspiracy theory" that should be acknowledged here (see Husting and Orr; Dentith, Husting and Orr; Harambam and Aupers, etc.). This literature suggests that people know their theories will be pejoratively labelled as "conspiracy theories" and it looks at the strategies deployed as a result, or the social consequences of that. Anyone aware of that literature will note that this section elides most of the really interesting discussion here, and thus this section should be revised in light of that. This is especially important because, as this affects how some people might criticise the studies this paper engages in. The considerations and criticisms levied above mean that it is not clear that the conceptualisation of studies 1 and 2 are quite as clean as the paper makes out.

For one thing, it is asserted that "If the motivated reasoning hypothesis is true, then participants will be motivated to take additional time to create justifications for why the conspiracy theories they believe are not conspiracy theories." Why assume this? Why not assume that the readers weren't, for example, trying to work out whether the theory is true, or how well this fits in with their other beliefs? This assumption a) seems arbitrary (it needs to be argue for), and b) it rests upon the assumption that the reader thinks anything labeled as a "conspiracy theory" is likely an irrational belief, and that is, as mentioned above, questioned within the literature. If people are taking their time over a theory labeled or implied to be a "conspiracy theory" and thinking "Should I endorse this or not?" then it is not clear that it is motivated reasoning that is causing delays. It might just be that people are more careful when thinking about conspiracy theories, or are cautious about endorsing them given public opprobrium towards such theories. It does not help that the motivated reasoning account is described again, later in the paper, in a different way.

This problem seems all the more weighty for the second study, as some participants where given a value-neutral definition, yet the paper seems to assume the irrationality of the theories in question because the author of the paper knows which ones they themselves think are true or false. Take the "COVID-19 (“the coronavirus”) was created in a lab in China as a bioweapon" claim; this series of claims is endorsed by various security and intelligence agencies but scorned by medical professionals. We might be able to have a debate as to which set of experts we should listen to, but for the general public their attitude as to how plausible this theory will be might well differ from those of us engaged in conspiracy theory theory research. Or take the "Jeffery Epstein was assassinated to prevent him from sharing information that would harm powerful politicians": I know of researchers in the field of studying conspiracy theory theory who think this is plausible to at least consider (which is not the same as saying it is "Completely true" but such a researcher might mark off on a Likert scales something close to the "Completely true" end).

Now, admittedly, the studies described in this paper indicate that conspiracy blindness, rather than motivated reasoning, is doing most of the work here. But a) the setup as to why we should even take the motivated reasoning account into account is scantly described or motivated and b) we are told that "The results of both Studies 1 and 2 were more consistent with the conspiracy blindness hypothesis than the motivated reasoning hypothesis." This "more consistent with" claim needs to be treated cautiously, because we are also told that some of the results in studies 1 and 2 are inconsistent with the conspiracy blindness hypothesis. What we seem to have here are results that are being interpreted with respect to two hypotheses, one of which is at least poorly motivated/described, and with results not fitting one hypothesis being seen as better fitting the other. Yet the mismatch between the operating (and non-pejorative) definition and the assumption that despite that definition, conspiracy theories really are irrational, seems to be doing a lot of the work when it comes to interpreting the results of these two studies.

In short, I would like to see this paper either grapple with using a non-pejorative definition but a pejorative framework, or mark out that the analysis in the paper is testing some pejorative assumptions in the existing literature. I would also like to see more motivation for the motivating reasoning account, as well as incorporating existing work on how labels might inhibit how people talk about conspiracy theories (including their own), especially since this paper seems to accept one aspect of the labelling debate: that if something is labelled as a "conspiracy theory" it probably is irrational to believe.

6. PLOS authors have the option to publish the peer review history of their article (what does this mean?). If published, this will include your full peer review and any attached files.

Reviewer #1: No

Reviewer #2: No

---

## [Author Response · Author response to Decision Letter 0]

1 Mar 2024

Manuscript PONE-D-23-41311

Response to Reviewers

Dear Dr. Gesser-Edelsburg,

Thank you for giving me the opportunity to revise my manuscript- and thank you to the reviewers who provided invaluable feedback. Your efforts are much appreciated. I have revised the manuscript accordingly, and I think it is all the better for it. As per PLOS ONE’s system, I am attaching the revised manuscript with and without tracked changes for easy viewing. 

I have included a point-by-point response to the reviewers below. The comments are organized by reviewer. My responses are labeled “response.” 

Reviewer Comments

Reviewer #1: This is an excellent study. A few minor comments:

1. I think the point could be brought out more that people believe propositions because they think those propositions are true. Since the term conspiracy theory often implies to people that the idea is not true, rarely will people label their own ideas as conspiracy theories. Usually, people like to label other people's ideas a conspiracy theories.

Response: This is absolutely true. I’ve adjusted the description of the conspiracy blindness hypothesis to clarify that this is one of the heuristics people may use that prevents them from considering that their own beliefs might be conspiracy theories (“This is true, conspiracy theories are false, and therefore this can’t be a conspiracy theory.”). (Top of Page 7) 

2. Some of the citations were a bit dated and there is some literature that might help make your case. You should cite:

Thalmann, Katharina. 2019. The Stigmatization of Conspiracy Theory since the 1950s:" A Plot to Make Us Look Foolish". New York: Routledge.

Douglas, Karen, Jan-Willem van Prooijen, and Robbie M. Sutton. 2022. "Is the Label ‘Conspiracy Theory’ a Cause or a Consequence of Disbelief in Alternative Narratives?" British Journal of Psychology n/a.

Dentith, M. R. X., G. Husting, and M. Orr. 2023. "Does the Phrase “Conspiracy Theory” Matter?" Society.

Response: Thank you so much for these suggestions! They are absolutely relevant here (particularly the Douglas, van Prooijen, & Sutton, 2022 paper), and I have added them to the manuscript. (Top of Page 6 and various other spots.)

3. The writingon the front end could be tightened.

The Heaton quote can go.

Most people believe one or a few conspiracy theories; Oliver and Wood only got 55% because they only asked about 7 cts. If you ask about more cts on a survey, more people believe at least one.

Cts are associated with harms; avoid making the causal argument: Uscinski, Joseph, Adam M. Enders, Casey Klofstad, and Justin Stoler. 2022. "Cause and Effect: On the Antecedents and Consequences of Conspiracy Theory Beliefs." Current Opinion in Psychology 101364. You could just cite the Jolley, Douglas, and Mari piece for consequences too.

Response: I have removed the Heaton quote and attempted to tighten up the writing in the introduction. (Page 1) 

You are correct that the true prevalence of conspiracy belief is likely higher than 55%. I have clarified this on page 3 and added an additional reference to historical Gallup polls showing belief in JFK conspiracy theories peaking at as high as 81% in the US to back this up.

Thank you! You are right- I made a mistake by using language that implied causality. Thank you for pointing this out! (Page 4)

4. The normative point should be made in the final discussion section: If an idea qualifies as a a conspiracy theory, does that mean that belief in it is unwarranted? The literature suggests yes Keeley 1999; Levy 2207; Uscinski and Enders 2023. At the same time, being a conspriacy theory does not mean false; rather it just means not proven in some way.

Response: Thank you for pointing this out! Based on this and Reviewer 2’s suggestions, I’ve clarified that not all belief in conspiracy theories is unwarranted throughout the manuscript. (For example, Page 3)

Reviewer #2: This is an interesting paper but I have some worries about the conceptualisation of both the theoretical framework the paper relies upon, and both of the studies.

Let me start with the theoretical framework. The paper assumes that what philosopher Charles Pigden calls a "modern superstition" about conspiracy theories is right - these are theories which are irrational. This is a bit of a problem for the paper, for two reasons: 1. The paper works with a fairly non-pejorative or evaluatively-laden definition of conspiracy theory ("conspiracy theories are explanations for events or circumstances that claim a group of powerful people is working together to accomplish a goal that comes at the expense of others, while attempting to keep their actions or intentions a secret") which admits in true conspiracies as being the subject of conspiracy theories; thus some of the presentation of material in the paper is at odds with itself when it assumes conspiracy theories are bad despite working with a definition that takes no sides. 

Response: Thank you for raising this concern! It was never my intention to argue that conspiracy theories are irrational- only that many lay people (and some scholars) perceive them as irrational. I wholly and unequivocally agree that belief in conspiracy theories can be completely rational depending on the context and the personal experiences of the believer. 

I was attempting to establish why someone might be motivated to reject the label of “conspiracy theorist,” but it seems I went too far. I have revised various parts of the paper to attempt to clarify this and prevent future misunderstandings.

2. The paper claims "These theories, like the people who believe them, are often considered irrational (McKenzie-McHarg & Fredheim, 2017; Wood & Douglas, 2013), even by the people who study them (Basham & Dentith, 2016; Bjerg & Presskorn-Thygesen, 2017)" - this is a problem because this is a misreading of Basham and Dentith, and probably not a kind reading of Bjerg & Presskorn-Thygesen. These authors have, to varying degrees, argued that the idea conspiracy theories and conspiracy theorists are irrational doesn't hold water when you examine said theories or said theorists.

Response: I heartily agree that that is what those authors were saying! I had cited them there as examples of scholars who argued that other academics were pathologizing (Basham & Dentith, 2016) or dismissing (Bjerg & Presskorn-Thygesen, 2017) conspiracy theories, and that it was inappropriate to do so. I see how my presentation could have been interpreted as these being examples of a phenomena rather than as critiques of that phenomena (as I’d intended). I have revised my phrasing on page 3 to clarify this.

The paper, as it stands, uses the "often" claim in a way that is only representative depending on what literature you admit or do not admit into the pool: in philosophy and sociology, for example, it is not considered prima facie irrational. I would recommend conditionalising this claim to say that in psychology and some of the social sciences the theories and the people who believe them are considered to be irrational, although in other fields of study around conspiracy theories, this claim is hotly contested. See the most recent special issue of Social Epistemology on this topic for details: https://www.tandfonline.com/toc/tsep20/37/4

Response: I had intended the “often” to apply to lay people, but not necessarily to (all) academics. As you’ve observed, people in many disciplines no longer treat conspiracy belief as irrational as they once did, and many scholars never did treat them that way. I see how my phrasing was misleading. I have updated my language on page 3 to attempt to clarify this. 

This kind of thing is important, especially given the evident misreading of at least one article cited in this paper. It is laudable that the paper works with a non-pejorative definition of conspiracy theory, and so it is important that if the paper wants to pursue an analysis of conspiracy theories under the lens that they are irrational, then the paper needs to point out that despite the value neutral definition some scholars who study conspiracy theories treat them as prima facie or largely irrational. The "some" here is important, as the irrationality of conspiracy theory belief generally is, it turns out, contested.

I also want to push back on another aspect of the pejorative tilt this paper takes: we are told "Belief in conspiracy theories can have serious consequences for public health, the environment, and democracy" and given a list of negative social consequences. But even scholars like Karen Douglas and Robbie Sutton have admitted that there are positive social consequences to belief in conspiracy theories: such as detecting conspiracies (on the definition proposed by this paper Woodward and Bernstein believed a conspiracy theory about the Watergate Affair, one that turned out to be true). Once again, the paper seems to be accepting at face value what merely some of the literature says about conspiracy theories.

Response: I have updated the introduction (on page 3) to include additional references to true conspiracies that began as conspiracy theories (like Watergate and MK Ultra), as well as some other potential benefits of belief in conspiracy theories.

This brings me to the short section on the motivating reason hypothesis, and it's slight discussion on labelling practices: there is already quite a lot of literature on the label "conspiracy theory" that should be acknowledged here (see Husting and Orr; Dentith, Husting and Orr; Harambam and Aupers, etc.). This literature suggests that people know their theories will be pejoratively labelled as "conspiracy theories" and it looks at the strategies deployed as a result, or the social consequences of that. Anyone aware of that literature will note that this section elides most of the really interesting discussion here, and thus this section should be revised in light of that. This is especially important because, as this affects how some people might criticise the studies this paper engages in. The considerations and criticisms levied above mean that it is not clear that the conceptualisation of studies 1 and 2 are quite as clean as the paper makes out.

Response: Thank you for pointing this out and for the recommendations for relevant literature! This is the argument that I was trying to make, but it seems that it did not come across. I’ve attempted to clarify this by explicitly discussing how people might be motivated to avoid the stigma associated with “conspiracy theories” to the motivated reasoning hypothesis. (Top of Page 6)

For one thing, it is asserted that "If the motivated reasoning hypothesis is true, then participants will be motivated to take additional time to create justifications for why the conspiracy theories they believe are not conspiracy theories." Why assume this? Why not assume that the readers weren't, for example, trying to work out whether the theory is true, or how well this fits in with their other beliefs? This assumption a) seems arbitrary (it needs to be argue for), and b) it rests upon the assumption that the reader thinks anything labeled as a "conspiracy theory" is likely an irrational belief, and that is, as mentioned above, questioned within the literature. If people are taking their time over a theory labeled or implied to be a "conspiracy theory" and thinking "Should I endorse this or not?" then it is not clear that it is motivated reasoning that is causing delays. It might just be that people are more careful when thinking about conspiracy theories, or are cautious about endorsing them given public opprobrium towards such theories. It does not help that the motivated reasoning account is described again, later in the paper, in a different way.

Response: I think that this argument assumes that lay beliefs about conspiracy theories and academic beliefs about conspiracy theories are the same. While I agree that conspiracy theories are not inherently irrational (as do many other academics), that does not seem to be the general perception among lay people (see the literature cited on page 3). Whether conspiracy theories are actually rational or irrational is irrelevant to the motivated reasoning hypothesis: All that matters is whether the person making judgments views them as irrational and/or stigmatized.

To address your second point, I have elaborated on the motivated reasoning hypothesis in the introduction (on page 6) that social desirability concerns (fear of judgment or fear of being seen endorsing a conspiracy theory) may also drive motivated reasoning in this case. In other words, people may be motivated to conclude that their beliefs do not belong to a socially stigmatized category.

There is, of course, the possibility that participants are outright lying about whether they recognize the conspiracy theories they believe as conspiracy theories. Fortunately, there is consistent evidence that people are more honest about stigmatized identities and behaviors and less concerned with social desirability in anonymous surveys like the ones in this paper (Dodou & Winter, 2014; Harenberg et al., 2022; Ong & Weiss, 2000). Unfortunately, they are still not completely honest (Ong & Weiss, 2000) so I cannot completely eliminate this possibility.

I am somewhat reassured, though, by that past research on honesty in anonymous surveys and the fact that the results are not consistent with that explanation. 

I’ve tried to make the wording of the motivated reasoning hypothesis more consistent throughout the paper. 

This problem seems all the more weighty for the second study, as some participants where given a value-neutral definition, yet the paper seems to assume the irrationality of the theories in question because the author of the paper knows which ones they themselves think are true or false. Take the "COVID-19 (“the coronavirus”) was created in a lab in China as a bioweapon" claim; this series of claims is endorsed by various security and intelligence agencies but scorned by medical professionals. We might be able to have a debate as to which set of experts we should listen to, but for the general public their attitude as to how plausible this theory will be might well differ from those of us engaged in conspiracy theory theory research. Or take the "Jeffery Epstein was assassinated to prevent him from sharing information that would harm powerful politicians": I know of researchers in the field of studying conspiracy theory theory who think this is plausible to at least consider (which is not the same as saying it is "Completely true" but such a researcher might mark off on a Likert scales something close to the "Completely true" end).

Response: The conspiracy theories selected for both studies are not necessarily false. I would be one of the researchers that you mention on the “believe” side of the Likert scale for the Epstein theory. I have clarified that these theories are not necessarily false or irrational- Just that they were not widely accepted as the mainstream narrative at the time the study was run. (Page 11) 

Now, admittedly, the studies described in this paper indicate that conspiracy blindness, rather than motivated reasoning, is doing most of the work here. But a) the setup as to why we should even take the motivated reasoning account into account is scantly described or motivated and b) we are told that "The results of both Studies 1 and 2 were more consistent with the conspiracy blindness hypothesis than the motivated reasoning hypothesis." This "more consistent with" claim needs to be treated cautiously, because we are also told that some of the results in studies 1 and 2 are inconsistent with the conspiracy blindness hypothesis.

Response: I have attempted to elaborate more on the motivated reasoning hypothesis in the introduction (Pages 5-6), and have been more cautious in describing the results of the studies. That said, I would argue that the most important results are completely consistent with the conspiracy blindness hypothesis, and the inconsistencies do not necessary contradict the conspiracy blindness explanation.

What we seem to have here are results that are being interpreted with respect to two hypotheses, one of which is at least poorly motivated/described, and with results not fitting one hypothesis being seen as better fitting the other. Yet the mismatch between the operating (and non-pejorative) definition and the assumption that despite that definition, conspiracy theories really are irrational, seems to be doing a lot of the work when it comes to interpreting the results of these two studies.

In short, I would like to see this paper either grapple with using a non-pejorative definition but a pejorative framework, or mark out that the analysis in the paper is testing some pejorative assumptions in the existing literature. I would also like to see more motivation for the motivating reasoning account, as well as incorporating existing work on how labels might inhibit how people talk about conspiracy theories (including their own), especially since this paper seems to accept one aspect of the labelling debate: that if something is labelled as a "conspiracy theory" it probably is irrational to believe.

Response: Overall, it seems that I tried too hard to argue that lay people often perceive conspiracy theories as irrational, leading to the misunderstanding that I was working under the assumption that all conspiracy theories are inherently irrational. I’d attempted to signal my stance by putting “irrational” in quotation marks in the original version of the manuscript and using a non-pejorative definition, but I see now that was too little. 

I am sincerely grateful for your honest feedback. I would have been very upset with myself if readers misjudged my position due to my subtlety. I am glad that you pointed this out to me before this manuscript went any further so I could head off any further misunderstandings. Thank you, thank you, thank you!

I hope that my revisions to the manuscript clarify my stance, and that this addresses the bulk of your concerns with this manuscript and its theoretical foundation. 

References

Dodou, D., & de Winter, J. C. (2014). Social desirability is the same in offline, online, and paper surveys: A meta-analysis. Computers in Human Behavior, 36, 487-495.

Harenberg, S., Ouellet-Pizer, C., Nieto, M., Kuo, L., Vosloo, J., Keenan, L., & Wilson, S. (2022). Anonymous vs. non-anonymous administration of depression scales in elite athletes: a meta-analysis. International Review of Sport and Exercise Psychology, 1-21.

Ong, A. D., & Weiss, D. J. (2000). The impact of anonymity on responses to sensitive questions 1. Journal of Applied Social Psychology, 30(8), 1691-1708.

---

## [Decision Letter · Decision Letter 1]

19 Mar 2024

Call it a Conspiracy: How Conspiracy Belief Predicts Recognition of Conspiracy Theories

PONE-D-23-41311R1

Dear Dr. Prims,

We’re pleased to inform you that your manuscript has been judged scientifically suitable for publication and will be formally accepted for publication once it meets all outstanding technical requirements.

Kind regards,

Anat Gesser-Edelsburg, Ph.D.

Academic Editor

PLOS ONE

Additional Editor Comments (optional):

Reviewers' comments:

Reviewer's Responses to Questions

**Comments to the Author**

1. If the authors have adequately addressed your comments raised in a previous round of review and you feel that this manuscript is now acceptable for publication, you may indicate that here to bypass the “Comments to the Author” section, enter your conflict of interest statement in the “Confidential to Editor” section, and submit your "Accept" recommendation.

Reviewer #1: All comments have been addressed

Reviewer #2: All comments have been addressed

2. Is the manuscript technically sound, and do the data support the conclusions?

Reviewer #1: Yes

Reviewer #2: Yes

3. Has the statistical analysis been performed appropriately and rigorously? 

Reviewer #1: Yes

Reviewer #2: Yes

4. Have the authors made all data underlying the findings in their manuscript fully available?

Reviewer #1: Yes

Reviewer #2: Yes

5. Is the manuscript presented in an intelligible fashion and written in standard English?

Reviewer #1: Yes

Reviewer #2: Yes

6. Review Comments to the Author

Reviewer #1: (No Response)

Reviewer #2: I am happy with the changes made to this paper. I also appreciate the author's humility when discussing instances where their intended meaning was opaque, thus leading to some of my and the other reviewer's criticisms. This kind of humility is to be applauded. I have no major worries about this paper now; my only minor concern is that more literature from the social scientists and philosophers who work with non-pejorative definitions could be cited, but this is really just a matter of personal preference.

7. PLOS authors have the option to publish the peer review history of their article (what does this mean?). If published, this will include your full peer review and any attached files.

Reviewer #1: No

Reviewer #2: **Yes: **M R. X. Dentith

---

## [Editor Report · Acceptance letter]

26 Mar 2024

PONE-D-23-41311R1 

PLOS ONE

Dear Dr. Prims, 

I'm pleased to inform you that your manuscript has been deemed suitable for publication in PLOS ONE. Congratulations! Your manuscript is now being handed over to our production team.

Kind regards, 

on behalf of

Prof. Anat Gesser-Edelsburg 

Academic Editor

PLOS ONE